# Geochronology, Geochemistry, and Pb–Hf Isotopic Composition of Mineralization-Related Magmatic Rocks in the Erdaohezi Pb–Zn Polymetallic Deposit, Great Xing'an Range, Northeast China

**Zhitao Xu** , **Jinggui Sun \*, Xiaolong Liang, Zhikai Xu and Xiaolei Chu**

College of Earth Sciences, Jilin University, Changchun 130061, China; xuzhjtao@163.com (Z.X.);
liangxl20@163.com (X.L.); xuzk18@163.com (Z.X.); chuxiaolei19@163.com (X.C.)
\* Correspondence: sunjinggui@jlu.edu.cn; Tel.: +86-431-88502278

**Abstract:** Late Mesozoic intermediate–felsic volcanics and hypabyssal intrusions are common across the western slope of the Great Xing'an Range (GXAR). Spatiotemporally, these hypabyssal intrusions are closely associated with epithermal Pb–Zn polymetallic deposits. However, few studies have investigated the petrogenesis, contributions and constraints of these Pb–Zn polymetallic mineralization-related intrusions. Therefore, we examine the representative Erdaohezi deposit and show that these mineralization-related hypabyssal intrusions are composed of quartz porphyry and andesite porphyry with concordant zircon U–Pb ages of 160.3 ± 1.4 Ma and 133.9 ± 0.9 Ma, respectively. These intrusions are peraluminous and high-K calc-alkaline or shoshonitic with high $Na_2O + K_2O$ contents, enrichment in large ion lithophile elements (LILEs; e.g., Rb, Th, and U), and depletion in high field strength elements (HFSEs; e.g., Nb, Ta, Zr, and Hf), similar to continental arc intrusions. The zircon εHf(t) values range from 3.1 to 8.0, and the $^{176}Hf/^{177}Hf$ values range from 0.282780 to 0.282886, with Hf-based Mesoproterozoic $T_{DM2}$ ages. No differences exist in the Pb isotope ratios among the quartz porphyry, andesite porphyry and ore body sulfide minerals. Detailed elemental and isotopic data imply that the quartz porphyry originated from a mixture of lower crust and newly underplated basaltic crust, while the andesite porphyry formed from the partial melting of Mesoproterozoic lower crust with the minor input of mantle materials. Furthermore, a magmatic–hydrothermal origin is favored for the Pb–Zn polymetallic mineralization in the Erdaohezi deposit. Integrating new and published tectonic evolution data, we suggest that the polymetallic mineralization-related magmatism in the Erdaohezi deposit occurred in a back-arc extensional environment at ~133 Ma in response to the rollback of the Paleo-Pacific Plate.

**Keywords:** zircon U–Pb dating; Hf and Pb isotopes; major and trace element geochemistry; erdaohezi Pb–Zn polymetallic deposit; Great Xing'an Range

## 1. Introduction

The Great Xing'an Range (GXAR) in Northeast China is an important part of the Central Asian Orogenic Belt (CAOB) and is characterized by numerous late Mesozoic volcanics and intrusions [1–6]. Furthermore, it is also one of the major global producers of polymetallic Pb and Zn resources; the Erguna metallogenic belt in the (GXAR) is a world-famous Pb–Zn polymetallic metallogenic belt [7–11] that hosts various types of metalliferous deposits (such as porphyries, hydrothermal veins, and epithermal deposits). The mineralization began during the late Mesozoic and peaked during 145–128 Ma [12–14] and occurred during intervals of volcanic activity or in the late stages of volcanic eruption, and is most closely related to intermediate–felsic intrusions.

The Erdaohezi deposit, which was discovered in 1980, is an epithermal Pb–Zn deposit in the the Erguna metallogenic belt (Figure 1A,B) [15], it is spatially associated with subvolcanic intrusions. Compared with other epithermal deposits in the study area, such as the Biliya, Derbur, Jiawula, Chaganbulagen and Erentaolegai deposits [16–20], the Erdaohezi deposit has been the focus of only a few studies. Moreover, few studies that have been performed on the petrogenesis, geochronology, major and trace element geochemistry, and Pb and Hf isotopic characteristics of the intermediate–felsic intrusions.

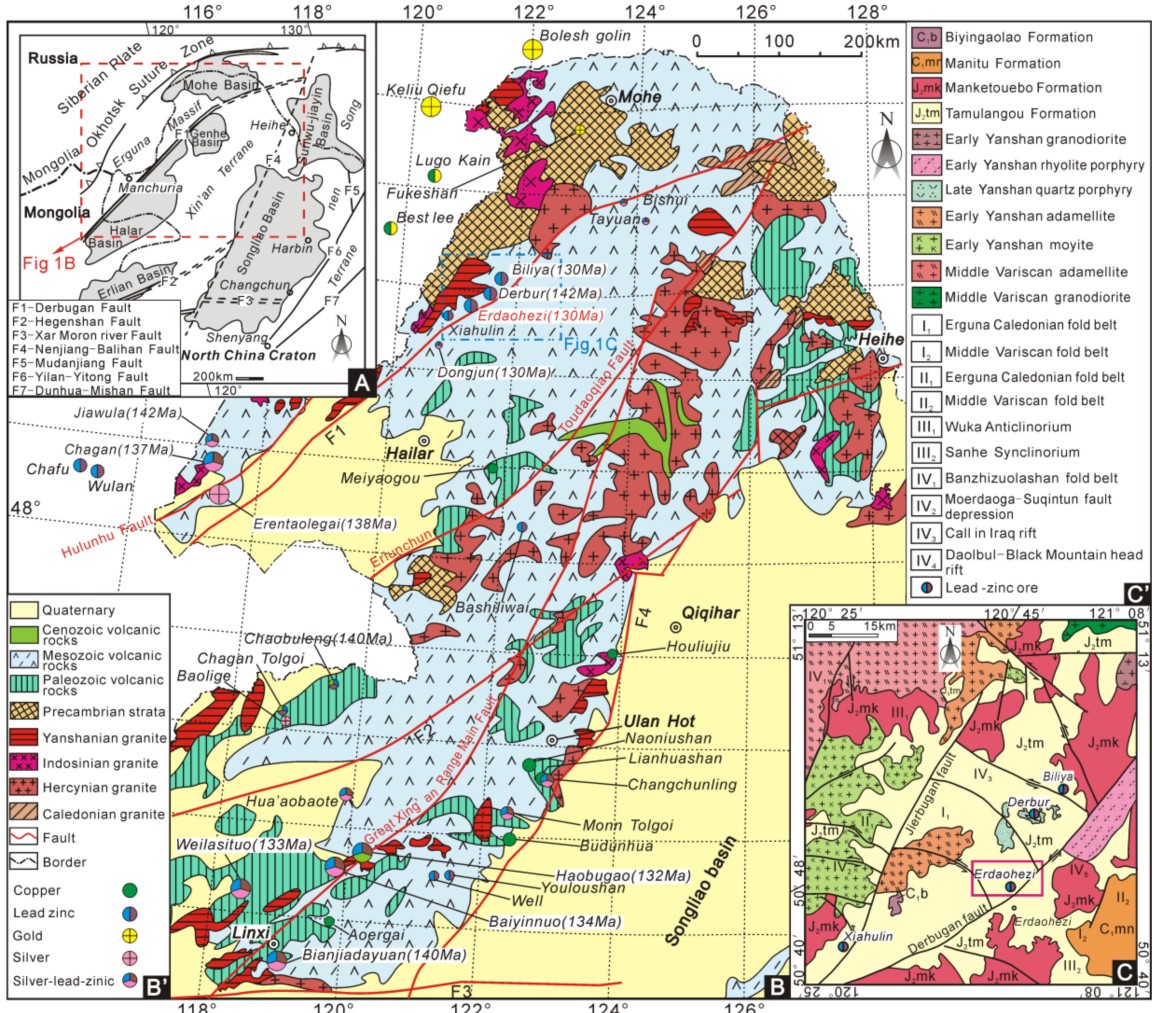

**Figure 1.** (**A**) Regional geological structure map of Northeast China [21]. (**B**) Geologic map and Pb–Zn deposits in the region [22]. (**C**) Regional geological map of the study area [23].

This study presents new geochronological age constraints from zircon U–Pb dating, Pb–Hf isotopic compositions and petrogeochemical data from andesite porphyry and quartz porphyry in the Erdaohezi deposit. Furthermore, we discuss the timing and petrogenesis of the quartz porphyry and andesite porphyry and establish the origin and evolution of magmatism and their relationships with Pb–Zn polymetallic mineralization in the region. Utilizing the regional geological evolution and spatiotemporal distributions of volcanic–subvolcanic rocks and Pb–Zn polymetallic deposits in the study area, we propose a possible geodynamic setting for the widespread intermediate–felsic magmatism and mineralization in the region and adjacent areas during the late Mesozoic (167–130 Ma). These results may provide a scientific basis for the regional prospecting and exploration of Pb–Zn polymetallic minerals.

## 2. Regional Geology

The Erdaohezi deposit is located on the western slope of the northern GXAR (Figure 1A) [24] within the northern part of the Erguna massif and is situated to the northwest of the Derbugan fault (Figure 2). Several northeast–southwest-striking faults divide this region into several blocks, including the Erguna massif, the Xing'an terrane, the Songliao basin, the Halar basin, and the Songnen terrane (Figure 1A) [25]. In this region, Early–Middle Jurassic magmatic events are correlated with the closure of the Mongol-Okhotsk Ocean and the formation of the Xing'an Mongolia Orogenic Belt (XMOB). Furthermore, the subduction of the oceanic Paleo-Pacific Plate may have been responsible for the large-scale intermediate–felsic magmatism and associated Pb–Zn polymetallic mineralization in the region [26–30].

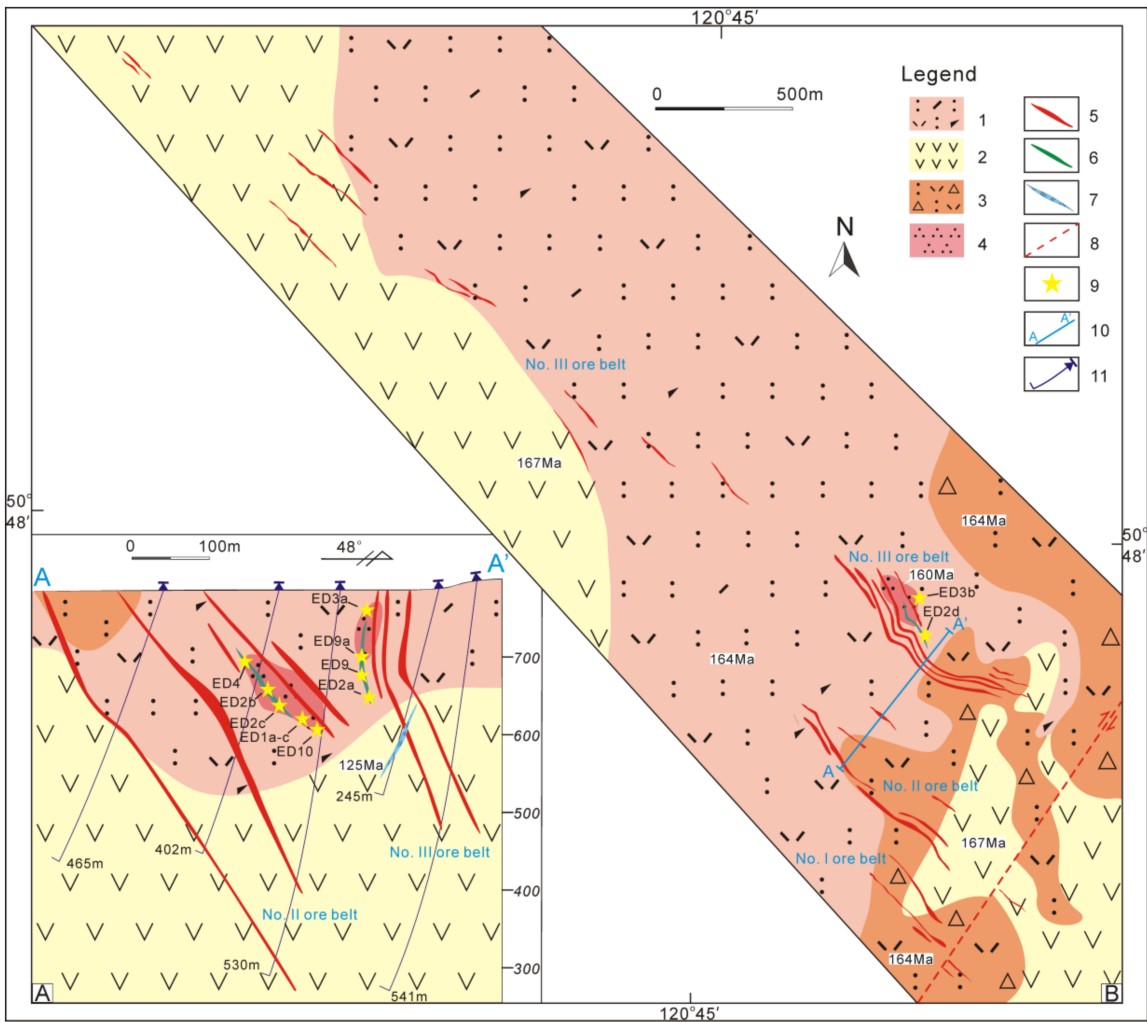

**Figure 2.** (**A**) Geological section of the Erdaohezi deposit. (**B**) Geological map of the Erdaohezi mining area [22]. 1. Rhyolitic crystal–lithic tuffs; 2. breccia; 3. volcanic rocks of the Tamulangou Formation; 4. rhyolitic tuffs; 5. quartz porphyry; 6. ore body; 7. andesite porphyry; 8. monzonite porphyry; 9. tectonic fracture zone; 10. fault; 11. sampling location; 12. cross section line; 13. borehole.

The evolution of regional tectonism in the study area has involved a variety of processes, namely the southward subduction of the Siberian Plate, the closure of the Okhotsk Ocean, and the long-range effects of the subduction of the Pacific Plate beneath Northeast China [26,31–36]. Within the Erguna region of the GXAR, the evolution of the geological structure can be briefly summarized as follows: (1) the Paleo-Asian Ocean closed completely, and the Mongol-Okhotsk back-arc basin formed (~259 Ma);

(2) the Paleo-Asian Ocean closed, and the Paleo-Pacific Plate began to subduct (~259–247 Ma); (3) the westward advance of the Paleo-Pacific Plate dominated the geotectonic environment coincident with the southward subduction of the Mongol-Okhotsk Ocean (~190–130 Ma); and (4) the Paleo-Pacific Plate retreated eastward, resulting in an extensional setting (with regional thinning or delamination of the lithosphere) in the Erguna massif (~130 Ma) [37–40]. Many different types of intrusions and hydrothermal deposits formed during the above-mentioned tectonic events in the region [41].

The exposed Mesoproterozoic basement rocks are mainly metamorphic rocks, including intermediate amphibolite, metamorphic gneiss [42,43], schist, quartzite and marble (1266 ± 3 Ma) [44]. The cap rocks are composed predominantly of Cambrian–Ordovician clastic rocks with shallow marine facies [45] as well as felsic, intermediate, and mafic volcanic rocks (159.2–166 Ma; [46,47]. The intrusions include primarily granodiorite–monzogranite (320–305 Ma) and diorite–granodiorite–granite (205–200 Ma) (Figure 1B,C). Late Paleozoic mafic–ultramafic rocks are developed mostly along the faults separating adjacent blocks [48–51]. The late Mesozoic magmatic rocks in the Erguna belt show a close spatiotemporal relationship with Ag–Pb–Zn–Cu mineralization, and the intrusions were emplaced in a shallow crustal setting as mid-hypabyssal, hypabyssal and ultrahypabyssal facies with felsic to intermediate compositions. The main rock assemblages include andesite porphyry [52], dacite porphyry, granite porphyry and quartz porphyry.

## 3. Geology of the Erdaohezi Deposit

The Erdaohezi deposit, located southwest of Genhe city, is a large epithermal Pb–Zn polymetallic deposit. Geological surveying of the mining area has revealed that the stratigraphy is dominated by Middle Jurassic felsic volcanics (rhyolitic lithic–crystal tuffs) of the Manketou Obo Formation (164 Ma) and basic–intermediate volcanic rocks (basaltic andesite) of the Tamulangou Formation (167 Ma). The ore bodies occur in the Tamulangou Formation volcanics, Manketou Obo Formation volcanics and quartz porphyry. Northwest and north-northwest-trending faults diverge to the northwest and converge to the southeast, sharing a genetic relationship with the volcanic edifice [53], and the spatial distribution of the Pb–Zn polymetallic bodies in the mining area is controlled by these faults. The hypabyssal intrusions in the Erdaohezi deposit are composed of Late Jurassic quartz porphyry and of Early Cretaceous andesite porphyry and monzonite porphyry. The monzonite porphyry formed after mineralization and thus cuts through the ore body, indicating the existence of multistage magmatic activity within the ore district. The mineralization of the Erdaohezi Pb–Zn polymetallic deposit is related to the quartz porphyry and andesite porphyry (Figure 2A,B; Figure 3A).

More than three mineralization belts are present in the Erdaohezi mining area; among them, the No. III ore belt is the main mineralization belt. The No. III ore belt contains 38 ore bodies, most of which occur as veins with dips of approximately 60–85° to the northeast and strikes of approximately 280–320° (Figure 2A). Veins of these ore bodies occur in the quartz porphyry intrusions in fracture zones or along the formation boundaries, and the occurrence of these ore bodies, which present primarily as veins, vein networks, and breccia structures, is consistent with that of the andesite porphyry. The main ore minerals are pyrite, sphalerite, galena, chalcopyrite, argentite and tetrahedrite, and small amounts of supergene oxides composed of limonite and covellite are present. Gangue minerals are quartz, fluorite, calcite, opal, sericite and chlorite, and rare adularia is observed.

The alteration zone of the surrounding rock can be divided into quartz–sericite ± illite–adularia (core), quartz–opal–calcite (middle) and fluorite–chlorite (margin). Three hydrothermal mineralization stages can be recognized: (I) gray quartz–pyrite–tawny sphalerite; (II) grayish white quartz–polymetallic sulfides; and (III) white quartz–pyrite.

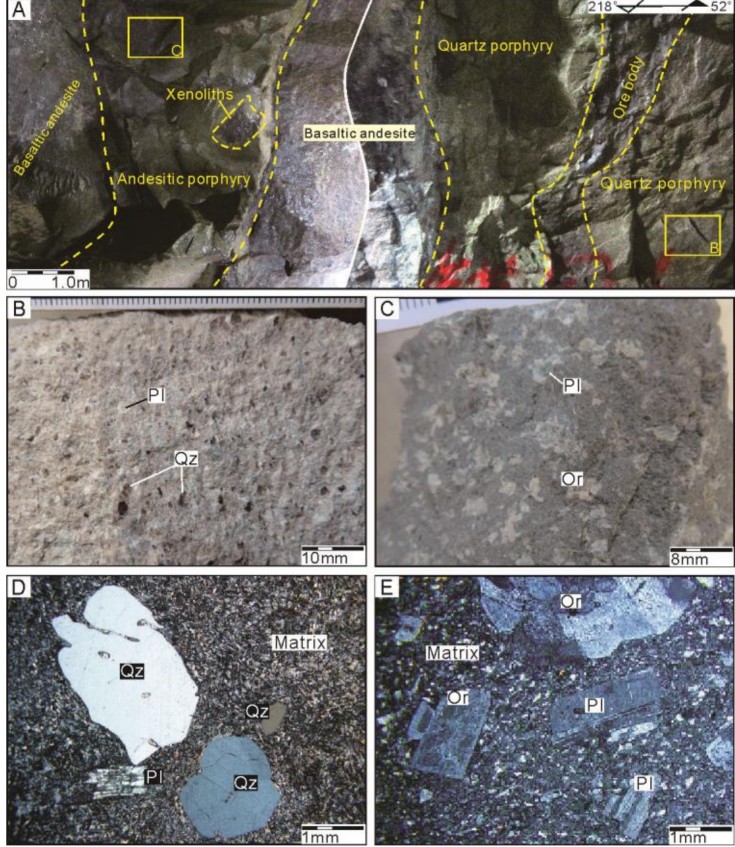

**Figure 3.** (**A**) Quartz porphyry cut by an ore vein and quartz porphyry cut by andesite porphyry; (**B**) hand specimen of quartz porphyry; (**C**) gray white quartz porphyry exposed in the field; (**D**) quartz porphyry showing mainly quartz phenocrysts and very few orthoclase and plagioclase phenocrysts; (**E**) andesite porphyry showing mainly plagioclase phenocrysts and very few orthoclase phenocrysts. Or–orthoclase; Pl– plagioclase; Qz–quartz.

## 4. Sampling and Analytical Techniques

### 4.1. Sample Descriptions

The metallogenesis-related intrusions in the Erdaohezi deposit include andesite porphyry and quartz porphyry. Representative samples (see Figure 2 for the sample localities, 120°45′11″–120°45′05″, 50°48′12″–50°48′08″) were selected for zircon dating (two samples, 28 zircon measurement points) and for whole-rock (two samples, 13 measurement points) and Pb–Hf isotope (two Pb samples and 29 Hf measurement points) geochemical analyses. The geological and petrographic characteristics of these rocks are described below and are shown in Table S1.

The quartz porphyry features a grayish white color and a porphyritic texture in the hand specimen (Figure 3A,B). The phenocrysts in the quartz porphyry ($1.0 \times 1.5$ mm$^2$ to $1.0 \times 2.0$ mm$^2$) are composed mainly of quartz (40–45 vol %), orthoclase (0–5 vol %) and plagioclase (10–15 vol %). Some plagioclase grains are variably sericitized. Quartz, orthoclase and plagioclase (<0.1 mm) can be identified in the matrix (30–35 vol %) (Figure 3D), and accessory minerals predominantly include zircon and altered mineral sericite.

The andesite porphyry occurs in the form of veins and features a light-gray color and porphyritic texture in the hand specimen (Figure 3A,C). The phenocrysts in the andesite porphyry ($0.6 \times 1.0$ mm$^2$ to $1.0 \times 1.2$ mm$^2$) are composed primarily of plagioclase (35–40 vol %), orthoclase (20–25 vol %), hornblende (5–10 vol %), and biotite (0–5 vol %). Orthoclase and plagioclase (<0.1 mm) can be identified

in the matrix (15–20 vol %) (Figure 3E). The mineral phenocrysts are euhedral to subhedral, and the accessory minerals include zircon, titanite, and others.

### 4.2. LA-ICP-MS Zircon U–Pb Dating

Zircons were selected from the quartz porphyry (ED10, 120°45′11″, 50°48′12″) and the andesite porphyry (ED4, 120°45′07″, 50°48′12″). First, using conventional heavy liquid and magnetic separation techniques, zircons were separated from the samples. Second, the samples were handpicked under a binocular microscope. Third, clear and euhedral zircon grains were mounted in epoxy resin. Fourth, the samples were polished to expose the zircon centers. The above work was completed at the mineral separation laboratory of Nanjing Hong Chuang Geological Exploration Technology Service Company Limited. All zircons were documented as reflected and transmitted light photomicrographs to reveal their internal structures and to select spots for in situ U–Pb dating and Lu–Hf analysis (Figure 4).

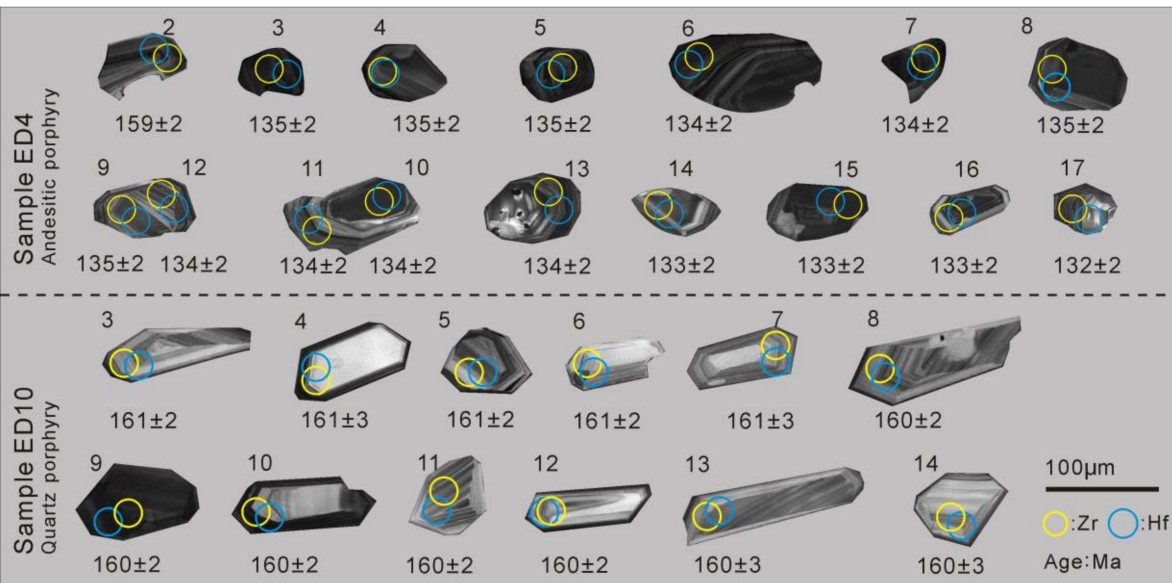

**Figure 4.** (**A**) Cathodoluminescence (CL) images of zircons from the quartz porphyry; (**B**) CL images of zircons from the andesite porphyry. The yellow circle represents the U–Pb age analytical spot, and the blue circle represents the Hf isotope analytical spot.

The zircon U–Pb isotopic compositions were analyzed at the State Key Laboratory for Mineral Deposits Research, Nanjing University (NJU). The samples were cleaned with water and weighed and then crushed, and the resulting powder was passed through 100–120 mesh equipment. Moreover, pure zircon crystals were selected under a binocular microscope. The above work was completed at the laboratory of the Regional Geological and Mineral Research Institute of Hebei Province. The zircon target and CL images were taken by Beijing Geoscience and Technology Company Limited and the Institute of Physics, Peking University, respectively. Laser ablation inductively coupled plasma mass spectrometry (LA-ICP-MS) U–Pb isotopic analyses were conducted using an Agilent 7500 mass spectrometer connected to a 193 nm ArF excimer laser ablation system [54,55]. First, a synthetic silicate glass standard reference material (NIST SMR610) was used to calibrate the instrument. Second, an international reference standard zircon (91500) was used as the external age calibration standard. The diameter of the laser spot was 32 μm, and the laser frequency was 10 Hz. Isotope ratios and element concentrations were calculated by Glitter software (ver. 4.4, Macquarie University). The age calculations and concordia plots by Isoplot (ver. 3.0) U–Pb fractionation were corrected using the GJ–1 [54], 91500 [56] and Mud Tank [57] zircon standards. The U–Pb ages were also calculated by Glitter software (ver. 4.4). The LA-ICP-MS U–Pb isotopic data are listed in Table S2. The U–Th–Pb age calculations were performed and the concordia diagrams were plotted using Isoplot/Ex ver. 3.0 [58].

### 4.3. Major and Trace Element Determinations

The geochemical sample analyses were performed at the Beijing Research Institute of Uranium Geology (BRIUG). The major element oxides were analyzed by a Panalytical PW4400 X-ray fluorescence spectrometer (analytical accuracy: 0.01 wt.%). Before the experiment, the GB/T14506.28 standard was used for $Na_2O$, MgO, $Al_2O_3$, $SiO_2$, $P_2O_5$, $K_2O$, CaO, $TiO_2$, MnO, and $Fe_2O_3$. The GB/T14506.28 standard was used for FeO, and the LY/T1253–1999 loss on ignition (LOI) standard was applied. Trace elements, including rare earth elements (REEs), were analyzed by an X-series plasma mass spectrometer (DZ/T0223–2001, analytical accuracy: 0.1 ppm). The major and trace element compositions of the quartz porphyry (ED10, ED3a–b, and ED1a–c) and andesite porphyry (ED4, ED9, ED9a, and ED2a–d) associated with the Erdaohezi mining area are given in Table S3.

### 4.4. Zircon Lu–Hf Isotopic Analyses

Zircon Lu–Hf isotopic analyses were conducted at the State Key Laboratory for Mineral Deposits Research, NJU, using a New Wave UP193FX laser ablation microprobe attached to a Neptune multi-collector ICP-MS [59,60]. In situ Lu–Hf isotopic analyses were performed predominantly with a beam diameter of 35 μm (repetition rate of 8 Hz). The 91500 standard zircon was used to monitor the performance conditions and analytical accuracy. The εHf values were calculated using a decay constant of $1.865 \times 10^{-11}$ per year [61,62]. Depleted mantle Hf model ages ($T_{DM2}$) were calculated using the measured $^{176}Lu/^{177}Hf$ ratios of zircon in reference to a depleted mantle model with a present-day $^{176}Hf/^{177}Hf$ ratio of 0.28325 [63], a $^{176}Lu/^{177}Hf$ ratio of 0.0384 [64] and an average continental crust ratio ($f_{CC}$) of −0.55 [64]. All Lu–Hf isotopic analysis results are reported with uncertainties of one standard deviation, and the results of the zircon Hf isotopic analyses performed in this study are given in Table S4.

### 4.5. Pb Isotopic Analyses

Whole-rock Pb isotopic measurements were conducted by a MAT–261 thermal ionization mass spectrometer. First, to remove surface contamination, approximately 50–80 mg of powder for each whole-rock sample was first leached in acetone. Second, the samples were washed with distilled water and dried at 60 °C in an oven. Third, each whole-rock sample was dissolved in distilled HF + $HNO_3$ (150 °C, 168 h). Fourth, the Pb was separated on Teflon columns using a HBr-HCl wash with an elution procedure and then loaded with a mixture of Sigel and $H_3PO_4$ onto a single Re filament (1300 °C). The measured Pb isotope ratios were corrected by repeated analyses of the NBS–981 Pb standard. The Pb isotopic measurements were better than two standard deviations, and the results of the whole-rock Pb isotopic analyses are given in Table S5.

## 5. Results

### 5.1. Whole-Rock Geochemistry

The quartz porphyry samples contain 74.87–76.36 wt.% $SiO_2$, 12.08–13.44 wt.% $Al_2O_3$, 3.56–3.93 wt.% $K_2O$, and 3.61–4.02 wt.% $Na_2O + K_2O$. In the $Na_2 + K_2O$ vs. $SiO_2$ (TAS) diagram, the three samples plot in the rhyolite field and are classified as subalkaline (Figure 5A) [65,66]. The Rittmann Index (δ) is 0.39–0.49, and the saturation index of aluminum (A/CNK) is 2.43–3.14, categorizing these rocks as high-potassium calc-alkaline and peraluminous rocks (Figure 5B) [67]. These values are similar to the major element compositions of the felsic rocks in the study area.

Compared with the quartz porphyry samples, the andesite porphyry samples have lower concentrations of $SiO_2$ (58.72–61.88 wt.%) and higher concentrations of $K_2O$ (3.89–4.56 wt.%) and $Al_2O_3$ (13.86–19.85 wt.%). In the TAS diagram (Figure 5A), the three samples plot in the andesite field and are classified as subalkaline. The δ and A/CNK values are 0.80–1.39 and 2.35–3.61, respectively, categorizing these samples as shoshonitic and peraluminous rocks (Figure 5B).

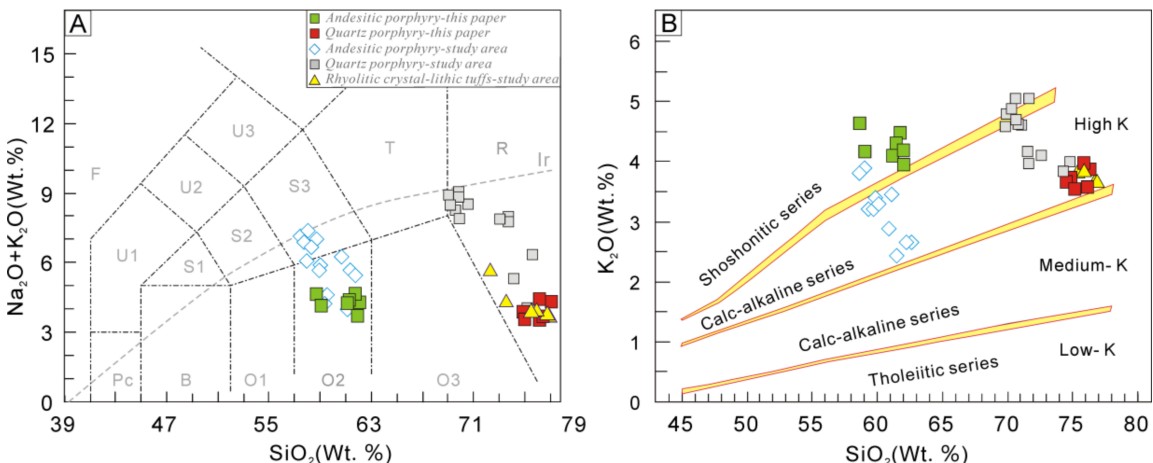

**Figure 5.** (**A**) $SiO_2$ versus ($Na_2O + K_2O$) diagram [66]. (**B**) $Si_2O$ versus $K_2O$ diagram [67]. B: basalt; O1: basaltic andesite; O2: andesite; O3: dacite; R: rhyolite; T: trachyte or trachydacite; S1: trachybasalt; S2: basaltic trachyandesite; S3: trachyandesite; Pc: picrobasalt; U1: basanite or tephrite; U2: phonotephrite; U3: tephriphonolite; F: foidite. Other data are quoted from the literature [68–71].

In Figure 6A (primitive mantle-normalized trace element spider diagram, [72]), the quartz porphyry samples are enriched in Th, U, and large ion lithophile elements (LILEs; e.g., Rb and K) and depleted in high field strength elements (HFSEs; e.g., Nb, Ta and Ti). Furthermore, the andesite porphyry samples feature lower contents of Th and U but higher contents of Ba, K, Ta and Nb than the host quartz porphyry samples (Figure 6C).

All the rocks have relatively low total REE contents ($\sum$REE = 99.35–190.12 ppm; Table S3). These patterns are characterized by light REE (LREE) enrichment and heavy REE (HREE) depletion (Figure 6B,D). The quartz porphyry samples are highly fractionated (LREE/HREE ratios ranging from 17.03 to 17.98 and $(La/Yb)_N$ values varying from 23.91 to 28.24 [73,74], with weakly negative Eu anomalies (0.47–0.68). In addition, the quartz porphyry is similar to the rhyolitic crystal–lithic tuffs in terms of the geochemical patterns of trace elements and REEs. In contrast, the andesite porphyry has a lower degree of fractionation (LREE/HREE values of 6.95–9.88 and $(La/Yb)_N$ values of 7.39–12.12) with smaller negative Eu anomalies (0.79–0.96). The Late Jurassic quartz porphyries and Early Cretaceous andesite porphyries in the study area (on the western slope of the GXAR) [75] are high-K calc-alkaline or shoshonitic rocks with similar contents of trace elements and REEs as the hypabyssal intrusions in the Erdaohezi deposit (Figure 6).

*5.2. Zircon U–Pb Dating*

The Erdaohezi quartz porphyry (ED10) and andesite porphyry (ED4) were selected for LA-ICP-MS zircon U–Pb dating (Table S2). Their zircon grains are euhedral–subhedral or prismatic (50–150 μm long) with aspect ratios of 1:1 to 4:1. Additionally, the zircons exhibit clear oscillatory zoning (Figure 4) and reveal U concentrations of 144–3054 ppm, Th concentrations of 152–3582 ppm, and Th/U ratios of 0.40–2.14, indicating that these zircons have a magmatic origin [76]. All of the analytical results plot near the concordia line (Figure 7).

Zircon grains collected from the quartz porphyry (ED10) yielded $^{206}Pb/^{238}U$ ages of 160–161 Ma with a weighted mean $^{206}Pb/^{238}U$ age of 160.3 ± 1.4 Ma (n = 12) (Figure 7A). Zircon grains collected from the andesite porphyry (ED4) yielded $^{206}Pb/^{238}U$ ages of 132–135 Ma with a weighted mean $^{206}Pb/^{238}U$ age of 133.9 ± 0.9 Ma (n = 15) (Figure 6B). Zircon ages of 159 Ma (Group I) correspond to zircons trapped in the quartz porphyry, which is consistent with geological observations made in the field.

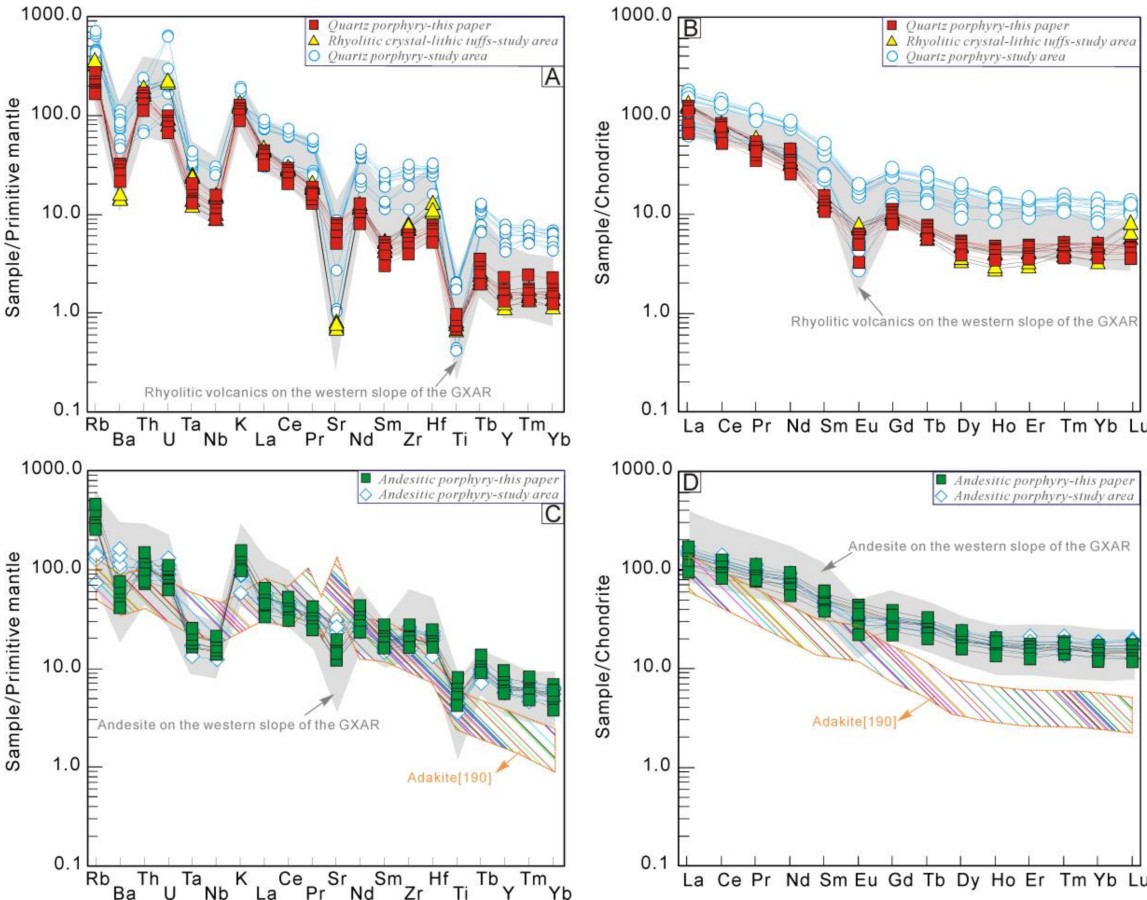

**Figure 6.** (**A**) Primitive mantle-normalized trace element spider diagrams of the quartz porphyry samples in the Erdaohezi deposit. (**B**) The chondrite-normalized rare earth element (REE) pattern of the quartz porphyry. (**C**) Primitive mantle-normalized trace element spider diagrams of the andesite porphyry samples in the Erdaohezi deposit. (**D**) The chondrite-normalized rare earth element (REE) pattern of the andesite porphyry. Other data are quoted from the literature [22,53,70,71,75].

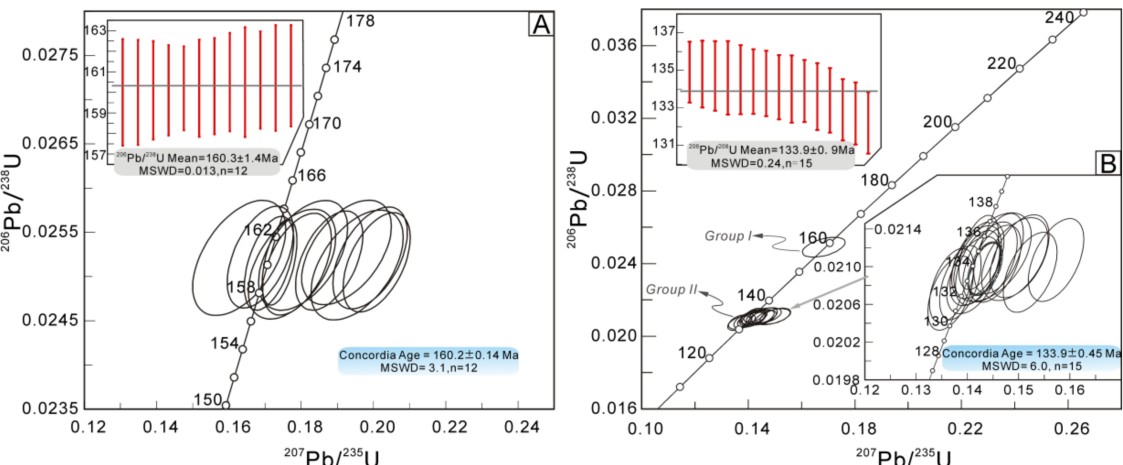

**Figure 7.** (**A**) Concordia diagrams of the zircon U–Pb ages for the quartz porphyry; (**B**) concordia diagrams of the zircon U–Pb ages for the andesite porphyry.

### 5.3. Zircon Lu–Hf Isotopic Data

The results of the Lu–Hf isotopic analyses of zircons from the quartz porphyry (ED10) and andesite porphyry (ED4) samples are shown in Table S4. The zircons from the quartz porphyry (160.3 ± 1.4 Ma) show a narrow range of Hf isotopic compositions, with $^{176}$Hf/$^{177}$Hf ratios ranging from 0.282847 to 0.282886. The εHf(t) values range from +5.7 to +8.0, and the $T_{DM2}$ ages range from 920 to 1130 Ma. The magmatic zircons from the andesite porphyry (133.9 ± 0.9 Ma) have $^{176}$Hf/$^{177}$Hf ratios ranging from 0.282854 to 0.282780 and εHf(t) values ranging from +3.1 to +5.8. The $T_{DM2}$ ages of the andesite porphyry range from 1106 to 1343 Ma, which are older than those of the quartz porphyry.

The analyzed zircons have Hf isotopic compositions that are similar to those of zircons from Phanerozoic intrusions elsewhere in the CAOB (Figure 8A) [77–80]. In addition, the analyzed zircons generally have relatively high $^{176}$Hf/$^{177}$Hf values, and all the sampled points fall between the fields of depleted mantle and young lower crust (above the 1.0 Ga line) (Figure 8B).

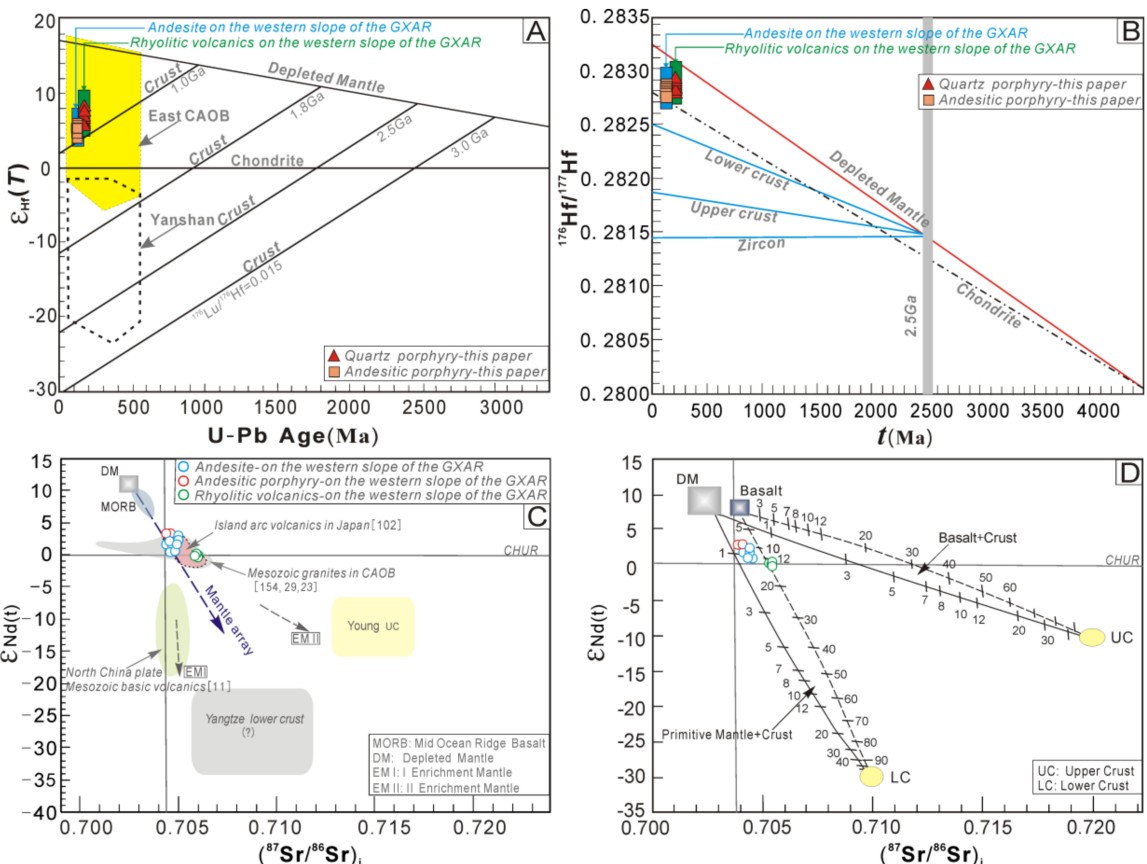

**Figure 8.** εHf(t) versus age plot of zircons (**A**) and $^{176}$Hf/$^{177}$Hf versus age plot of zircons (**B**) [81,82]. ($^{87}$Sr/$^{86}$Sr)$_i$–εNd(t) diagrams (**C**) and (**D**) of the andesite porphyry, andesite and rhyolitic volcanics on the western slope of the Great Xing'an Range (GXAR) [83]. Data for the andesite and rhyolitic rocks on the western slope of the GXAR are quoted from the literature [84–89].

### 5.4. Pb Isotopic Compositions

The Pb isotopic compositions of four samples measured in the Erdaohezi deposit and previous studies are listed in Table S5. The Pb isotopic compositions of the andesite porphyry and quartz porphyry in the study area [22] are characterized by minor variations in $^{206}$Pb/$^{204}$Pb, $^{207}$Pb/$^{204}$Pb and $^{208}$Pb/$^{204}$Pb (Figure 9A,B), with values ranging from 18.438 to 18.476 (2σ: 0.0004 to 0.0020), 15.571 to 15.622 (2σ: 0.0008 to 0.0020), and 38.224 to 38.263 (2σ: 0.0010 to 0.0040), respectively.

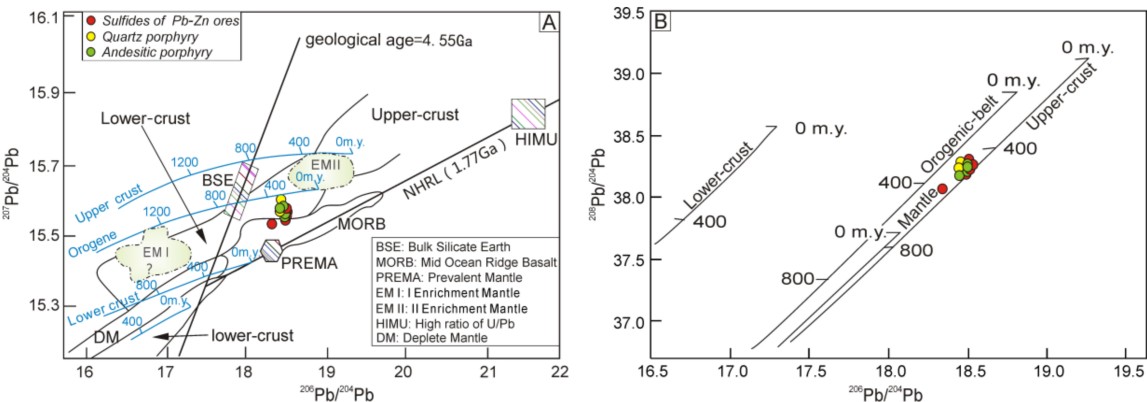

**Figure 9.** (**A**) $^{207}Pb/^{204}Pb$ versus $^{206}Pb/^{204}Pb$ diagram for the andesite porphyry, and (**B**) $^{208}Pb/^{204}Pb$ versus $^{206}Pb/^{204}Pb$ for the quartz porphyry in the Erdaohezi deposit (the base diagrams are from [88]).

## 6. Discussion

### 6.1. Timing of Magmatism and Mineralization

Previous studies on regional magmatism and metallogenesis in the study region concentrated mostly on particular epithermal Pb–Zn polymetallic deposits on the western slope of the GXAR, such as the Derbur [89], Dongjun [90], Jiawula, Chaganbulagen [91], Weilasituo and Bianjiadayuan deposits. As a result, large quantities of geochronological data have been obtained (Table S7); for example, from the Derbur, Dongjun, Jiawula, Chaganbulagen, Weilasituo, Bianjiadayuan, Baiyinnuo and Haobugao epithermal Pb–Zn polymetallic deposits, which formed primarily in the Early Cretaceous (Figure 1, Table S6) [92–100].

In this paper, we present new ages for the quartz porphyry and andesite porphyry of 160.3 ± 1.4 Ma and 133.9 ± 0.9 Ma, respectively. Moreover, on the basis of the ages of the felsic and intermediate intrusions related to Pb–Zn polymetallic mineralization in the Erguna metallogenic belt, we postulate that both types of intrusions formed primarily in the Late Jurassic or Early Cretaceous, with the felsic intrusions forming at 140–160 Ma and the intermediate intrusions forming at 133–148 Ma [101]. Thus, the ages of the quartz porphyry and andesite porphyry in the study area are similar to the ages of these intrusions.

Finally, considering the Rb–Sr isochron age (130.5 ± 3.6 Ma, n = 5; personal communication) obtained from ore body sulfides in the main metallogenic stage from the Erdaohezi deposit, the timing of emplacement of the andesite porphyry is consistent with the timing of mineralization, indicating a temporal relationship between these processes. Thus, the magmatism and mineralization of the Erdaohezi deposit occurred in the Early Cretaceous.

### 6.2. Petrogenesis and Nature of Magma Source

#### 6.2.1. Quartz Porphyry

Currently, from research conducted on the global scale, three main hypotheses have been proposed to explain the source of rhyolitic magma in this region: (1) the partial melting of crustal rocks [102–105], (2) the separation and crystallization of basaltic or andesitic magma [106–113], and (3) bimodal volcanism [108,109]. Furthermore, the late Mesozoic basaltic rocks on the western slope of the GXAR are less widespread than the rhyolitic rocks in the study area, and the basaltic rocks in the study area are either fine-grained or cryptocrystalline with no phenocrysts, indicating that a basaltic magma was rapidly erupted and quenched. Therefore, it is unlikely that a large volume of rhyolitic magma separated from the basaltic magma. Furthermore, the Rb–Sr age of the basalts in the Tamulangou Formation is approximately 155 Ma [114], which is later than the age of the rhyolitic volcanics in the Manketou Obo Formation (164 Ma).

In terms of the REE content, the rhyolitic rocks contain REEs in less abundance than the basaltic andesite and basalt; moreover, the Eu anomalies of the rhyolitic rocks are stronger, and their LREE/HREE values are higher. With regard to trace elements, the rhyolitic lithic–crystal tuffs are generally enriched in LILEs, depleted in HFSEs (Table S3), and especially depleted in Sr. The distribution of elements in these tuffs is different from that in the mafic-intermediate volcanics, but the REE characteristics of the rhyolitic lithic–crystal tuffs are similar to those of the felsic volcanics in the region (Figure 8A,B). Moreover, the $(^{87}Sr/^{86}Sr)_i$ and $\varepsilon Nd(t)$ values of the rhyolitic volcanics in the study area are quite different from those of the basaltic andesite and basalt on the western slope of the GXAR, suggesting that there is no evolutionary relationship between them (Figure 8C).

The rhyolitic lithic–crystal tuffs in the Erdaohezi deposit have high $SiO_2$ and low MgO contents and are categorized as high-K calc-alkaline rocks (Figure 5B); their Nb/Ta and Zr/Hf ratios are close to those of the crust, with a crust-derived parent magma [115,116]. Therefore, the rhyolitic rocks in the Manketou Obo Formation are probably the product of the partial melting of the lower crust. Furthermore, the diagenetic age of the rhyolitic rocks is similar to that of the basaltic andesite (167 Ma). Combined with the geochronological data obtained in this study, the timing of andesitic magmatism in the region was obviously later than that of the rhyolitic magmatism. We further postulate that the rhyolitic magma was not sourced from a basaltic magma or an andesitic magma. These characteristics of the mafic-intermediate and felsic volcanics are similar to those of the bimodal volcanics on the western slope of the GXAR, and the rhyolitic rocks and basaltic andesite in the Erdaohezi deposit can be classified as bimodal volcanics (Figure 11A).

In terms of the petrogeochemistry, the $K_2O/SiO_2$ ratio of the quartz porphyry (0.05) is similar to that of the rhyolitic rocks, and both rock types are classified as calc-alkaline rocks (Figure 5B). These rocks have similar major element ratios, such as $Al_2O_3/SiO_2$, $TiO_2/SiO_2$, and $Na_2O/K_2O$, and similar $\delta$ values (Table S3). They feature high $Ce/Al_2O_3$ ratios (110.18–1355.56) and low $TiO_2/Al_2O_3$ ratios (0.01–0.06), which are characteristic of subvolcanic rocks in a continental arc setting (Figure 10A). Their normalized REE patterns are generally similar and feature high LREE enrichment. The LREE/HREE and $\delta Eu$ values of both rock types are similar to those of the rhyolitic rocks and are characterized by high differentiation and negative $\delta Eu$ values. Additionally, the trace elements of the quartz porphyry are similar to those of the felsic volcanics (Figure 6A). Indeed, the Th vs. Th/Nd and $\varepsilon Nd(t)$ vs. $(^{87}Sr/^{86}Sr)_i$ diagrams (Figure 11C; Figure 8C,D) further indicate that a depleted mantle-derived magma partially melted the lower crust and became contaminated with crustal materials [117]. The Zr/Hf and Nb/Ta values of the quartz porphyry are similar to those of the rhyolitic lithic–crystal tuffs (Table S3), suggesting that their parent magmas were similar. These geochemical characteristics are similar to those of rhyolitic intrusions on the western slope of the GXAR and suggest that they may have been sourced from the partial melting of the crust (Figure 10B) and from subsequent mixing with a small amount of depleted mantle material or newly underplated basaltic crust.

The zircon Hf isotopic data from the Erdaohezi deposit are similar to the Hf isotope values of the quartz porphyry in the study region (Figure 8A). The quartz porphyry is characterized by positive $\varepsilon Hf(t)$ values (+5.7 to +8.0), and the $T_{DM2}$ model ages range from 920 Ma to 1130 Ma, indicating that the primary magmas may have originated from a depleted mantle or newly underplated basaltic crust (Figure 8B). Moreover, the ranges of the $(^{87}Sr/^{87}Sr)_i$ and $\varepsilon Nd(t)$ values in the rhyolitic rocks in the region are 0.70485–0.70619 and −0.1–1.4, respectively. These values indicate a mixed source region consisting of 85–88% newly underplated basalt and 12–15% lower crust in the Mesoproterozoic (Figure 8C,D). Similarly, the Pb–Pb isotopic data (Figure 9A,B) from the quartz porphyry in the Erdaohezi deposit support a mixed source consisting of newly underplated basaltic crust and lower crust in the Mesoproterozoic.

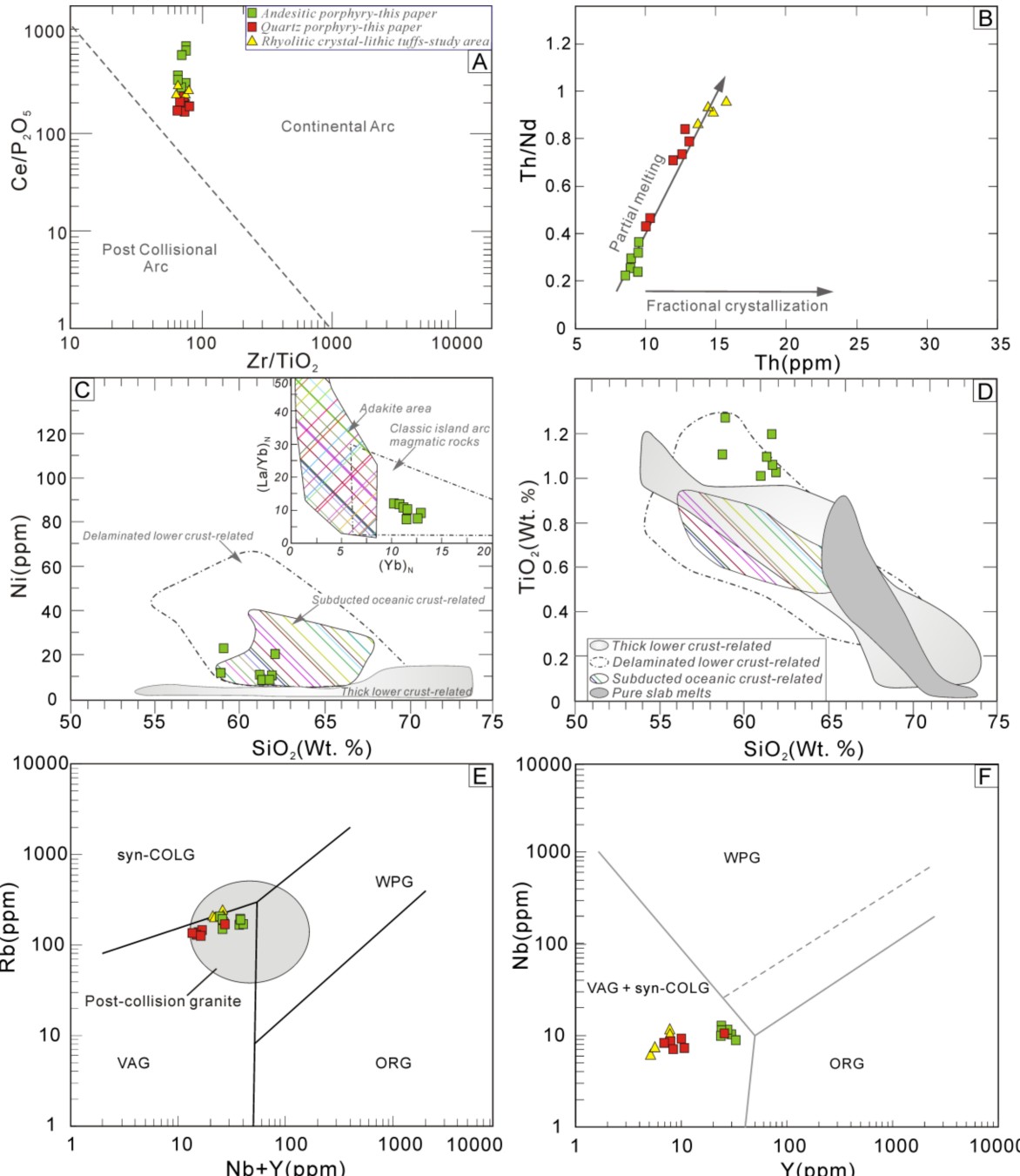

**Figure 10.** (**A**) Ce/P$_2$O$_5$ vs. Zr/TiO$_2$ diagram [118]. (**B**) Th vs. Th/Nd diagram [119,120]. (**C**) SiO$_2$ vs. Ni diagram [74]. (**D**) SiO$_2$ vs. TiO$_2$ diagram [74]. (**E**) Rb vs. Nb + Yb diagram. (**F**) Nb vs. Y plot [121,122]. Other data are from [22,53,70,71,75].

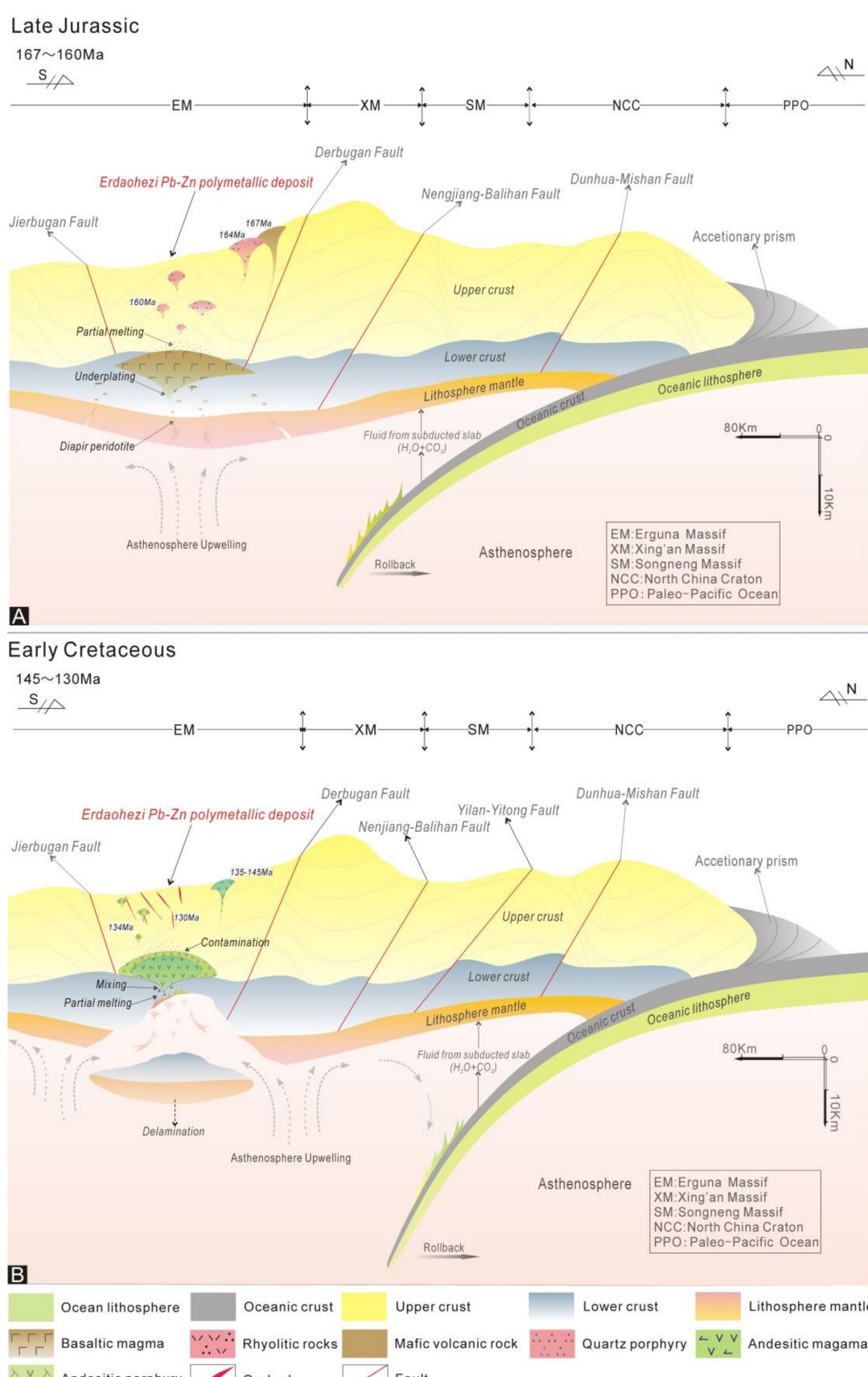

**Figure 11.** Genetic model and tectonic setting for the Late Jurassic quartz porphyry (**A**), and Early Cretaceous andesite porphyry (**B**) in the GXAR.

### 6.2.2. Andesite Porphyry

The age of the andesite porphyry obtained in this paper is 133.9 Ma, which is slightly younger than the age of the andesitic volcanics in the Manitu Formation within the study region (135–145 Ma) [123]. The following hypotheses have been proposed for the petrogenesis of the andesitic volcanics in the Manitu Formation on the western slope of the GXAR: (1) the partial melting of mafic rocks in the lower crust [124,125]; (2) the mixing and differentiation of mafic magma and felsic magma to varying degrees [126,127]; and (3) independent magma origins, including asthenospheric upwelling and partial melting of the lower crust or a subducted oceanic plate [128,129]. Previous studies have shown that the andesitic volcanics in the Manitu Formation are widespread across the western slope of the GXAR and that they include mainly andesite and a small amount of andesitic tuff. Moreover, there is no large-scale distribution of Early Cretaceous basaltic rocks on the western slope of the GXAR (Figure 1B,C), and the andesitic rocks in the Manitu Formation have a weak Eu anomaly. All of these results indicate that the andesitic rocks in the study area may not be related to the separation and crystallization of a basaltic magma, nor are they likely the product of the mixing of basaltic and rhyolitic magmas. In terms of their petrogeochemistry, the andesitic volcanics in the Manitu Formation are peraluminous, calc-alkaline shoshonites enriched in Rb, Th and U and depleted in Ta, Nb and Zr, without obvious Sr depletion (Figure 6C,D). Thus, their magma source may have been affected by mantle material or a subducted plate. Furthermore, their Rb/Sr, Th/U, Hf/Sm, Ta/Hf, Nb/Ta, Sm/Nd, and $La_N/Yb_N$ ratios are similar to those of continental arc andesites, and their $^{176}Hf/^{177}Hf$ and $\varepsilon Hf$ (t) values suggest that their diagenetic magma may have originated from the partial melting of the lower crust, following which the magma was contaminated with material from the lower crust (Figure 8A,B; Figure 11B).

Concerning the petrogeochemistry of the andesite porphyry in the study area, the $K_2O/SiO_2$ ratio (0.06–0.08) is similar to that in other andesites. All the andesite porphyry samples belong to the calc-alkaline or shoshonitic series (Figure 5B) and have similar ratios of major elements (such as $Al_2O_3/SiO_2$, $TiO_2/SiO_2$, and $Na_2O/K_2O$) and δ values (Table S3). Furthermore, the andesite porphyry has a high $Ce/Al_2O_3$ ratio (110.18–1355.56) and a low $TiO_2/Al_2O_3$ ratio (0.01–0.06), showing the same characteristics of continental arc rocks as the other andesites (Figure 10A). The normalized REE patterns are generally similar; the LREE/HREE and δEu values are similar and are characterized by weak differentiation and weakly negative δEu values [130]. The trace element compositions are enriched in Rb and K and depleted in Ba, Ta, Nb, Sr and Ti (Figure 6A,C), and the Th and Th/Nd values in the andesite porphyry in the Erdaohezi deposit are similar to those of the andesites on the western slope of the GXAR. The Th vs. Th/Nd and $\varepsilon Nd$(t) vs. $(^{87}Sr/^{86}Sr)_i$ diagrams (Figure 10A; Figure 8C,D) further indicate that the depleted mantle-derived parent magma partially melted the lower crust and subsequently became contaminated by crustal materials. The Zr/Hf and Nb/Ta values of the andesite porphyry (40.60–46.18 and 15.33–21.79, respectively) are similar to those of other andesites, further suggesting that the parent magma was characterized by crust–mantle mixing. Therefore, these rocks may have been sourced from the partial melting of the lower crust and from mixing with a small amount of mantle material. In summary, these rocks in the study region are similar to intrusions related to continental arcs.

The above geochemical characteristics are also supported by the zircon Hf isotopic data (Figure 8A,B). The $\varepsilon Hf$(t) values of the andesite porphyry range from +3.1 to +5.8, and the corresponding $T_{DM2}$ ages range from 1106 to 1343 Ma, indicating that a mixture of depleted mantle and lower crustal materials was involved in the petrogenesis of the andesite porphyry. Moreover, the ranges of the $(^{87}Sr/^{87}Sr)_i$ and $\varepsilon Nd$(t) values in the andesite porphyry are 0.70455–0.70496 and +1.9–+2.8, respectively; these values plot near the depleted mantle evolution line (Figure 8C) and suggest that the mixed source was composed of 82–88% depleted mantle and 8–12% lower crust in the Mesoproterozoic (Figure 8D). Compared with the rhyolitic magma, the andesitic magma may have contained more mantle-derived materials.

In addition, the Pb isotopic data from the andesite porphyry plot in a broadly similar field (Figure 9A,B); these results also support a mixed crust–mantle source. The Pb isotopic compositions

of the intrusions and sulfide minerals in the Erdaohezi deposit show broadly similar $^{206}Pb/^{204}Pb$, $^{207}Pb/^{204}Pb$, and $^{208}Pb/^{204}Pb$ ratios ranging from 18.308 to 18.475, 15.528 to 15.622, and 38.078 to 38.319, respectively. These values suggest that these rocks originated from a source composed of young lower crust with a small amount of depleted mantle material. Furthermore, the Pb isotopic compositions of the intrusions and sulfides clearly plot in the field between those of orogenic belts and mid-ocean ridge mantle (Figure 9A,B), possibly indicating some affinity to orogenesis related to the tectonic activity of the Paleo-Pacific Plate.

*6.3. Geodynamic Setting of Diagenesis and Metallization*

Numerous studies on the magmatic–hydrothermal deposits on the western slope of the GXAR have shown that most of the ore systems therein are related to local late Mesozoic magmatism [131–135]. In addition, available geochronological results illustrate that the majority of the magmatic–hydrothermal deposits formed between 160 and 130 Ma [136–142] (Table S6). Four models have been proposed for the magmatic–tectonic history in the study region: (1) the Rodinia supercontinental extensional faulting model [143–146]; (2) the continental margin model related to the tectonic activity of the Mongol-Okhotsk Ocean [147]; (3) the continental margin model related to the activity of the Pacific Plate [39]; and (4) the mantle plume underplating model [108].

The Late Jurassic to Early Cretaceous quartz porphyry (164–140 Ma) and Late Jurassic rhyolitic lithic–crystal tuffs have geochemical compositions that range from high-K calc-alkaline to shoshonite, whereas the andesite porphyry (148–133 Ma) exhibits an opposite trend (Figure 5B; Table S3). All these rocks show the petrogenetic characteristics of continental arc rocks [148]. In addition, the REE fractionation patterns are consistent (Figure 6B,D): the LREE/HREE and $La_N/Yb_N$ values of the quartz porphyry indicate strong differentiation, while those of the andesite porphyry indicate weak differentiation. Additionally, the Th/Yb, Ta/Yb and Nb/Yb ratios are indicative of continental arc rocks. In the plots of Rb versus Yb + Y and Nb versus Y, the sample points of the studied rocks plot within the postcollisional granite, collisional granite and volcanic arc granite fields, which are associated with suitable extensional environments related to the subduction and compression of the oceanic lithosphere beneath the Asian continental plate (Figure 10E,F). Moreover, the $SiO_2/Ni$ (2.53–7.11), $SiO_2/TiO_2$ (45.07–60.56), and $SiO_2/Cr$ (0.62–2.03) ratios of the andesite porphyry further suggest that the parent magma was related to delamination of the lower continental crust (Figure 10C,D).

Nevertheless, the petrogenesis and tectonic affiliation of the Late Jurassic volcanics and intrusions in the Erguna metallogenic belt on the western slope of the GXAR remain controversial. However, it is widely accepted that the activity of the Pacific Plate played a dominant role in the evolution of the late Mesozoic magma [149,150], the Late Jurassic magmatic episode, and the transition of the tectonic setting from compressional to extensional. Furthermore, a number of geochemical and geophysical studies have indicated that the late Mesozoic magmatism on the western slope of the GXAR occurred within different tectonic settings [151,152]. The Late Jurassic volcanics and intrusions likely formed within a continental arc, with a setting transitioning from compressional to extensional. In the Late Jurassic, the subduction of the Paleo-Pacific Plate reached its maximum extent with a low angle, where the slab extended more than 1300 km west to the Erguna massif on the western slope of the GXAR [153]. It has been suggested that the tectonic transformation in this region caused significant lithospheric thickening as a consequence of the subduction of the Pacific Plate [154–157]. Based on the timing of magmatism and polymetallic mineralization in the region, during 167–160 Ma, the subduction of the Paleo-Pacific Plate resulted in lithospheric thickening of the Erguna massif and formed a continental arc [158–162]. With the complete subduction of the Paleo-Pacific Plate and the weakening of the compressive stress, an extensional environment gradually formed subsequent to subduction, followed by the upwelling of asthenospheric material, resulting in the partial melting of the accreted basaltic lower crust and the formation of the rhyolitic magma. This process resulted in the emplacement of the magma associated with the quartz porphyry (160 Ma) in the study area. The regional rhyolitic and basaltic volcanics in the Tamulangou Formation constitute bimodal volcanic rocks. Based on

the Sr–Nd–Pb–Hf isotopic compositions of these rocks (Figure 8), we consider these volcanics to be the product of the partial melting of the Mesoproterozoic lower crust caused by asthenospheric mantle upwelling in the extensional environment that formed following the subduction of the oceanic Paleo-Pacific Plate. Moreover, the quartz porphyry exhibits the same geochemical characteristics as the rhyolitic lithic–crystal tuffs, although the emplacement timing of the former was slightly later than that of the latter, which indicates that the Late Jurassic quartz porphyry occurred late in the eruption process of the rhyolitic lithic–crystal tuffs (Figure 11A).

The Early Cretaceous (145–120 Ma) was an important time for intracontinental extension after the subduction of the Paleo-Pacific Plate [163,164]. Regional geophysical data [165,166] and the spatial distribution of Mesozoic magmatic rocks [167–169] suggest that the late Early Cretaceous magmatism that occurred throughout the western slope of the GXAR can be attributed to the aftereffects of the subduction of the Paleo-Pacific Plate. Additionally, this period also represents the peak of large-scale lithospheric thinning in Northeast China [170–173]. During this period, the thickened lower crust and lithospheric mantle in the study area began to delaminate due to gravitational instability, and this delamination caused the position of the oceanic plate to migrate gradually with respect to the position of the continental margin [174–177]. This delamination also caused the large-scale upwelling of asthenospheric mantle, which further promoted the change in the subduction angle of the oceanic plate from low to high. The upwelling of the asthenosphere increased the thermal gradient, caused the partial melting of the thickened lower crust and potentially provided the heat source responsible for the widespread intermediate–felsic magmatism in the study area. The Paleo-Pacific slab likely continued to experience rollback due to thermal upwelling, resulting in the upwelling of asthenospheric mantle, lithospheric thinning, delamination, and the formation of large-scale Pb–Zn polymetallic mineralization and the emplacement of the high-alumina shoshonitic andesite porphyry during the Early Cretaceous (Figure 11B). The diagenetic tectonic setting of these intrusions exhibits postorogenic extensional characteristics (Table S7), and this setting is further evidenced by the existence both of voluminous Early Cretaceous volcanics and intrusions and of the extensional basins in the GXAR.

## 7. Conclusions

The zircons from the quartz porphyry and andesite porphyry yielded U–Pb ages of 160.3 ± 1.4 Ma and 133.9 ± 0.9 Ma, respectively, indicating that the magmatism related to the mineralization in the Erdaohezi deposit possibly occurred in the Early Cretaceous. The quartz porphyry formed from a mixed source composed of Mesoproterozoic lower crust and newly underplated basaltic crust, and the andesite porphyry formed from the partial melting of Mesoproterozoic lower crust with the minor input of depleted mantle. Both porphyries formed in an extensional environment related to the Paleo-Pacific Plate at 133 Ma.

**Supplementary Materials:** The following are available online at http://www.mdpi.com/2075-163X/10/3/274/s1: Table S1: Geological and petrographic characteristics of andesitic porphyry and quartz porphyry from the Erdaohezi deposit, Table S2: LA-ICP-MS zircon U-Pb data of andesitic porphyry (ED4) and quartz porphyry (ED10) from the Erdaohezi lead-zinc deposit, Table S3: Whole-rocks geochemical data of the intrusions from the Erdaohezi deposit(major element: wt%; trace elements: ppm), Table S4: In situ or para position Hf isotopic analyses of zircons from the Erdaohezi deposit, Table S5: Lead Isotopic compositions of the Erdaohezi lead-zinc deposit, Table S6: Compilation of intrusions and lead-zinc polymetallic mineralization ages in Great Xing'an Range, Table S7: The classification and main features of the low-sulfidation, medium-sulfidation and high-sulfidation epithermal deposits.

**Author Contributions:** Conceptualization, Z.X. and J.S.; methodology, J.S.; software, X.L.; validation, Z.X., J.S., X.L., Z.X. and X.C.; formal analysis, X.C.; investigation, Z.X., J.S. and X.L.; resources, J.S.; data curation, Z.X. and J.S.; writing—original draft preparation, Z.X.; writing—review and editing, Z.X. and J.S.; visualization, Z.X.; supervision, X.C.; project administration, J.S.; funding acquisition, J.S. All authors have read and agreed to the published version of the manuscript.

**Funding:** This work is funded by the National Natural Science Foundation of China (41172072) and the Geological Survey of China (DD20160344).

**Conflicts of Interest:** The authors declare no conflict of interest.

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
