# Peer review of "Geochronology, Geochemistry, and Pb–Hf Isotopic Composition of Mineralization-Related Magmatic Rocks in the Erdaohezi Pb–Zn Polymetallic Deposit, Great Xing’an Range, Northeast China"

_minerals, doi:10.3390/min10030274_

Round 1
Reviewer 1 Report
I have work through the revised manuscript of Xu et al. and have found this to be well suited for publication now. All issues I raised in my original review have been taken care off.
I have made slight corrections to table 2 (attached), otherwise these are fine as well.

Author Response
Dear, Editor-in-Chief and reviewer,
We received your letter detailing the reviewers’ comments on the manuscript, and read each of the suggestions carefully. The comments by the reviewers and editor were very novel, and we believe that they significantly helped improve the manuscript. With the help of our tutor, our scientific research team, we have made the following revisions to the manuscript:
R1: We have made slight corrections to table 2
To facilitate the manuscript review by the editor and reviewers, we used red font to mark the revised portions of the manuscript.
Thank you very much for the editor's and peer reviewers' comments on our manuscript. We look forward to hearing from you.
Sincerely,
Dr. Jing-gui Sun and Dr. Zhi-tao Xu
Jinlin University
Zhi-Tao Xu and Jing-Gui Sun
College of Earth Sciences, Jilin University, Changchun, China
Postal address: No. 2199 Jianshe Street, Changchun 130061, Jilin Province, China
Tel: (0086)13844175339
Email address: sunjinggui@jlu.edu.cn

Reviewer 2 Report
Report on the manuscript entitled: "Geochronology, geochemistry, and Pb–Hf isotopic composition of mineralization-related magmatic rocks in the Erdaohezi lead–zinc polymetallic deposit, Great Xing’an Range, NE China”,
by: Zhi–Tao Xua, Jing–Gui Suna, Xiao–Long Lianga, Zhi–Kai Xua and Xiao–Lei Chua.
Ms number: minerals-716147
Comments from Reviewer
Overview
The ms by Xu et al. et al. concerns Pb-Zn polymetallic mineralization-related intrusions from the Erdaohezi deposit. Data presented here associate petrography-mineralogy, geochemical (major and trace elements) and Hf-Pb isotopes data combined with U-Pb geochronology on two quartz porphyry and andesitic porphyry hypabyssal intrusions which display concordant zircon U–Pb ages of 160.3 ± 1.4 Ma and 133.9±0.9 Ma, respectively. A magmatic-hydrothermal origin is favored by the authors for the Erdaohezi Pb-Zn mineralization. It is suggested that the polymetallic mineralization-related magmatism occurred in a back-arc extensional environment at ~133 Ma in response to the rollback of the Paleo-Pacific Plate.
The ms is well-written, clear, well-organized (except section 5, see below) and the general plan of the ms is suitable. The figures are all necessary and of very good quality. The reference list is (too) long but seems to be complete and up-to-date even I don’t know (I don’t have access to) the papers from, for example, Acta Petrologica Sinica or Journal of Jilin University.
Data presented by Xu and co-authors reach a high scientific level in any of the addressed topics and the interpretations are reliable. There is no doubt that this work will serve as a reference for the understanding of the regional geology but also of the genesis of magmatic hydrothermal mineralization in their geodynamic framework. Therefore, this ms can be accepted for publication with minor revisions.
Specific comments or details
- The key-word Element geochemistry is too vague, too general. Major and trace elements geochemistry is better.
- In the whole text replace acidic by felsic
- The section 5 “Results” must be re-organized. Indeed, the 5.2 whole rock chemistry section must be moved to 5.1 and therefore appear before 5.1 Zircon U-Pb dating section. This will make the text more fluid, relevant.
- The discussion section seems to me too long and the relevant ideas of the authors are drowned in the midst of too many references from articles often repeated several times in the same paragraph. This section should therefore be simplified, lightened-shortened to make it more impactful.
Conclusion, I think that this manuscript can be accepted for publication in an international journal as Minerals with minor improvements in form.
Author Response
Dear, Editor-in-Chief and reviewer,
We received your letter detailing the reviewers’ comments on the manuscript, and read each of the suggestions carefully. The comments by the reviewers and editor were very novel, and we believe that they significantly helped improve the manuscript. With the help of our tutor, our scientific research team, we have made the following revisions to the manuscript:
- The key-word Element geochemistry is too vague, too general. Major and trace elements geochemistry is better.
Re: This is a good idea. We have replaced “Element geochemistry” with “Major and trace elements geochemistry”.
- In the whole text replace acidic by felsic
Re: Yes, we have corrected this problem.
- The section 5 “Results” must be re-organized. Indeed, the 5.2 whole rock chemistry section must be moved to 5.1 and therefore appear before 5.1 Zircon U-Pb dating section. This will make the text more fluid, relevant.
Re: This is a novel proposal. We have reorganized the section 5.
- The discussion section seems to me too long and the relevant ideas of the authors are drowned in the midst of too many references from articles often repeated several times in the same paragraph. This section should therefore be simplified, lightened-shortened to make it more impactful.
Re: According to the reviewer's suggestion, in the 6st part of the discussion, we have shortened this part, such as avoiding repeated references, deleting redundant words and simplifying sentences.
To facilitate the manuscript review by the editor and reviewers, we used red font to mark the revised portions of the manuscript.
Thank you very much for the editor's and peer reviewers' comments on our manuscript. We look forward to hearing from you.
Sincerely,
Dr. Jing-gui Sun and Dr. Zhi-tao Xu
Jinlin University
Zhi-Tao Xu and Jing-Gui Sun
College of Earth Sciences, Jilin University, Changchun, China
Postal address: No. 2199 Jianshe Street, Changchun 130061, Jilin Province, China
Tel: (0086)13844175339
Email address: sunjinggui@jlu.edu.cn

This manuscript is a resubmission of an earlier submission. The following is a list of the peer review reports and author responses from that submission.
Round 1
Reviewer 1 Report
This article is not highly original but addresses an interesting topic and provides a needed documentation of an epithermal environment. In addition to the comments formulated below, my main comments are: 1) shortening, there is a lot in this article that could be move to tables, graphics, or that are digressions not essential to the issue at ends; and 2) make sure that you are not comparing apples and potatoes when it comes to comparing the chemistry of different intrusions (I suggest comparing sources compositions, petrogenetic models and else, instead of comparing Sr… contents).
English – something needs to be done about the English. The abstract is a good example – sentences like ‘as the research object and carry out these works.’ gives a bad impression, others are hard to understand, and the use of punctuation (; in particular in the abstract) needs to be corrected. Please hire a native English speaker to do this.
References – this article is not formatted as a typical MDPI contribution. Please use Mendeley, Endnote or else to correct this.
Title – I suggest reformulating: ‘Constraints of magmatism’ – not clear what is meant by that
Abstract – well constructed, informative. It is however too long, and the English needs to be corrected by a native speaker.
Introduction – well organised and informative. The problematic comes a bit late but, as some background is needed to understand why this study is important, it works. About the English, it is especially important to reformulate the problematic sentence (l. 69-71) – it is an extremely important sentence but it is hard to understand in its present form.
Fig. 2 – some work is needed on this figure to give it a publication-standard aspect.
195-198 – how many samples are we talking about? + add GPS coordinates (table) 228 – sure, but you still need to give us enough details so that we know exactly how these zircons have been analysed, without having to read another article (idem l.249) 336-etc. – what are the detection limits? QA-QC?... 390-397 – the article is too long and this is a good example of the type of data that could be integrated to a figure (graphic) – this would shorten the text and make that article easier to read 411 – I don’t know what to do with ‘under review’ data – I don’t think they should be mentioned in this text (or use a formulation like: M. …, personal communication)Section 6.2.1. – there is a lot of talk about bimodal magmatism in this section, but no explanation as to what the source of felsic melt is in that kind of model. Please add an explanation, and built your argument by telling us what chemical characteristics point to differentiation and which provide indications on the source. Also, all this talk about this has more K than this magma should be shortened (idem for l. 523-536)– please interpret the geochemistry in terms of what it means for the petrogenesis of your intrusions (it is much more powerful to compare petrogenetic models).
476-etc. – to reformulate, these sentences don’t make senseSection 6.3. – should be shorten, and all these information about ‘what is the metallogenic model?’ should be presented (succinctly) in the Geological Setting section (this is not an article on the metallogenic model). Then the comparison with other magmas should be modified or removed. We have no idea what is being compared (magmas or rocks with variable degrees of alteration?) and magma comparison should be done more carefully (at similar SiO2 content, for example). This section is not scientifically sound.
Fig. 12 – legend to be edited
631-648 – this could be shortened tooConclusion – a conclusion is needed, the text of l. 724-733 is not a conclusion
Reviewer 2 Report
The manuscript by Xu et al. reports a wealth of new isotope and geochemical data on the Erdaohezi lead–zinc polymetallic deposit in NE China. I find the data the be presented in a clear and consistent manner, the implications derived therefrom are sound and straight forward. I cannot comment on the overall geological findings and interpretations as I am no expert in the local geology. But the conclusions drawn from the data and the literature seem to be sound.
There are only a few issues with the text and data presentation which I have directly given in the manuscript and the supplement.
